# Friction Stir Spot Welding-Brazing of Al and Hot-Dip Aluminized Ti Alloy with Zn Interlayer

**Xingwen Zhou** [1,2], **Yuhua Chen** [1,*], **Shuhan Li** [1,*], **Yongde Huang** [1], **Kun Hao** [1] and **Peng Peng** [2]

1   School of Aerospace Manufacturing Engineering, Nanchang Hangkong University, Nanchang 330063, China; xingwenzhou@buaa.edu.cn (X.Z.); huangydhm@nchu.edu.cn (Y.H.); 13362624883@163.com (K.H.)

2   School of Mechanical Engineering and Automation, Beihang University, Beijing 100191, China; ppeng@buaa.edu.cn

*   Correspondence: ch.yu.hu@163.com (Y.C.); shuhanli@outlook.com (S.L.);
    Tel.: +86-133-3006-7995 (Y.C.); +86-156-7912-8826 (S.L.)

**Abstract:** Friction stir spot welding (FSSW) of Al to Ti alloys has broad applications in the aerospace and automobile industries, while its narrow joining area limits the improvement of mechanical properties of the joint. In the current study, an Al-coating was prepared on Ti6Al4V alloy by hot-dipping prior to joining, then a Zn interlayer was used during friction stir joining of as-coated Ti alloy to the 2014-Al alloy in a lap configuration to introduce a brazing zone out of the stir zone to increase the joining area. The microstructure of the joint was investigated, and the joint strength was compared with the traditional FSSW joint to confirm the advantages of this new process. Because of the increase of the joining area, the maximum fracture load of such joint is 110% higher than that of the traditional FSSW joint under the same welding parameters. The fracture load of these joints depends on the joining width, including the width of solid-state bonding region in stir zone and brazing region out of stir zone.

**Keywords:** friction stir spot welding; friction stir spot brazing; joining area; fracture load

## 1. Introduction

Lightweight hybrid structures have attracted increasing attention in the aerospace and automobile industries to enhance performance efficiency as well as to reduce environmental impact [1–4]. In this context, the dissimilar joining of aluminum (Al) and titanium (Ti) alloys has been widely used in the manufacturing of lightweight hybrid structures [4–6]. However, joining these two alloys is a considerable challenge because of their different physical properties, limited mutual solubility, and formation of brittle intermetallic phases in the Al–Ti alloying system [7–10].

Several solid-state joining processes, such as friction stir welding (FSW) [11]; diffusion bonding [12,13]; and fusion welding with filler metals, such as laser welding [10,14,15] and arc welding [8,16], were thus developed for dissimilar joining to overcome the difficulties mentioned above. Among these, an innovative solid-state spot welding process, friction stir spot welding (FSSW), is a promising method that can reduce the formation of intermetallic compounds (IMCs) because of its low processing temperature, and has various advantages, such as a simple process, as well as being environment-friendly and energy efficient [5,6,17]. In a typical FSSW, a rotating cylindrical pin tool is plunged into the workpiece to be welded. Frictional heat is generated in the plunging and stirring stages, and thus the materials adjacent to the tool are heated, softened, and mixed in the stirring stage where a solid-state joint will be formed [18]. Up to now, many reports on FSSW of similar or dissimilar metals mainly focus on the process, microstructural characteristics, and numerical simulations to

optimize the mechanical properties of the joint [17–21]. Unavoidably, the traditional FSSW would leave a keyhole that reduces its bonding area [22]. Although some modified processes for FSSW, such as refill FSSW [23], flat FSSW [24], and short travel FSSW [25], have attempted to eliminate this visible keyhole, they are complicated and time-consuming owing to the rigorous welding conditions and/or complex tool design. Recently, friction stir brazing (FSSB) has been developed by using a pinless tool and adding brazing filler metal to induce metallurgical reaction at the interface with the aid of friction heat and forging pressure, instead of plastic flow [22]. This technique has successfully introduced a brazing zone in dissimilar joints of steel/Al [26], Cu/Al [27], and Al /Mg [28], while FSSB of Al and Ti alloys is yet to be reported.

This work aims to find an effective approach to improve the joining area of the Al/Ti joint by combining FSSW and FSSB. Zinc (Zn) is an optional filler for FSSB of Al alloys as a result of its low melting point (420 °C), high solubility in Al, as well as absence of formation of IMCs in the Al–Zn alloying system [29–32]. During our preliminary study, a Zn interlayer was placed between the interface of 2014 Al and Ti6Al4V alloys for the FSSB process to determine their weldability. It was found that the high affinity of Ti towards oxygen led to the formation of non-protective oxide scales on the Ti6Al4V [33], resulting in poor wettability of Zn to Ti alloys and, therefore, failure to join Al and Ti alloys. Recently, using an electroplating process before the FSB process would produce a copper layer on the surface of graphite, which converted the graphite–copper lap joint to the similar joining of copper to copper [34]. This gives us an idea that pretreating the base metal will be a possible benefit for FSSB of Al and Ti.

In the present paper, we utilized hot dip to aluminize T6Al4V alloys and developed a friction stir spot joining process, called friction stir spot welding-brazing (FSSW-B), for joining of 2014-T4 Al and aluminized T6Al4V alloys with a Zn interlayer. The effect of tool rotation speed on the interfacial structure and mechanical properties of joints was investigated in detail and the relation between microstructure, mechanical behavior, and process parameters was established.

## 2. Experimental

The substrate materials under study were aluminum alloy 2014-T4 and annealed titanium alloy Ti6Al4V sheets, both 3-mm thick and with dimensions of approximately 80 mm × 35 mm. Table 1 lists the nominal chemical composition of the two alloys.

**Table 1.** Chemical compositions of the 2014-T4 aluminum alloy and Ti6Al4V titanium alloy (wt.%).

| Alloys | Cu | Si | Mn | Mg | Fe | Zn | Ti | Ni | Al | V | C | N | H | O |
|--------|------|-----|------|------|-------|------|------|------|------|-----|-------|-------|-------|------|
| 2014-T4 | 4.3 | 1.0 | 0.73 | 0.55 | 0.3 | 0.08 | 0.02 | 0.02 | Bal. | - | - | - | - | - |
| Ti6Al4V | - | - | - | - | 0.026 | - | Bal. | - | 6.0 | 4.0 | 0.015 | 0.008 | 0.007 | 0.06 |

A molten bath consisting of pure Al (99.9 wt.%) was used for the hot-dip aluminizing of the Ti6Al4V alloy. Prior to the hot-dip aluminizing, the specimens were treated with an acid pickling solution containing 10 vol.% HF, 10 vol.% HNO$_3$, and 80 vol.% H$_2$O for 10–15 s followed by washing in water and acetone, and then drying. The molten Al bath was kept at 780 °C, and the treated Ti6Al4V specimens were immersed in the bath for 25 min. The aluminized Ti6Al4V samples were subsequently removed from the molten bath and then dried in air. Figure 1a shows the cross-section micrograph of the hot-dipped Al coating on the Ti6Al4V substrate. The coating has a thickness of about 200 μm and dispersed particles that have an average diameter of 5.2 μm. These particles have an atomic ratio of Ti to Al of about 1:3 (composition of 22.4 Ti-77.6 Al (in at. %)), which is consistent with the composition of TiAl3. Furthermore, the interface between the alloy and the coating is not flat but undulating, and a continuous TiAl3 layer with a mean thickness of 7.3 μm formed at the interface (see Figure 1b). The content of TiAl3 in this Al coating could be controlled by varying the hot-dip aluminizing process [35,36].

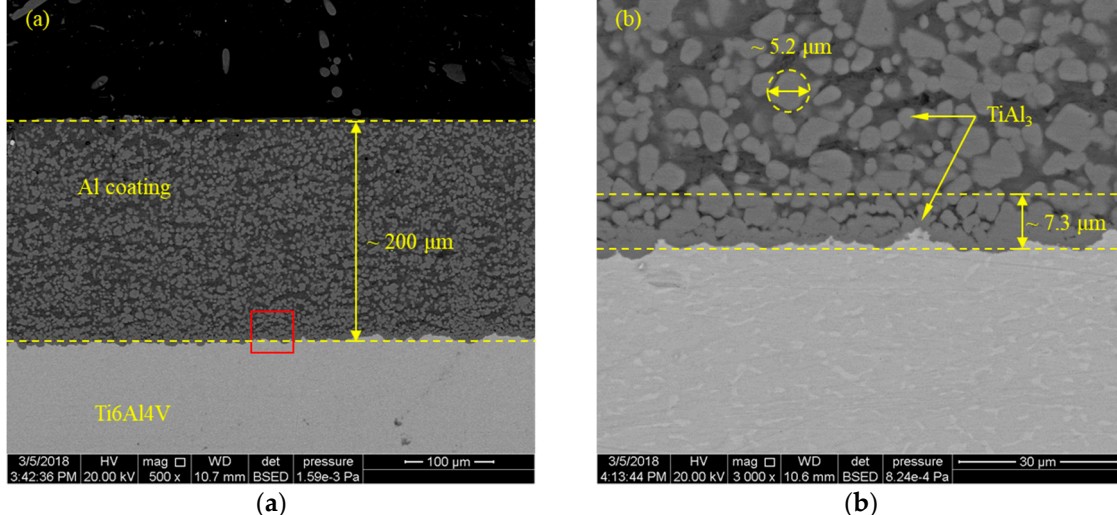

**Figure 1.** Cross-sectional morphology of the hot-dip aluminized Ti6Al4V alloy; (**a**) low magnification and (**b**) high magnification (red square marked in (**a**)).

Figure 2 shows the schematic illustration of the FSSW-B setup. All the FSSW-B operations were conducted in lap joint configuration so that the Al plate was always on top of the Ti sheet with a 35 mm × 35 mm overlap between two sheets. The welding machine used in this study was modified from an X53K milling machine (Tonmac, Nantong, China). The stir tool was made of a GH4169 superalloy, with a tool shoulder of 16 mm and a circle threaded pin 5 mm in diameter. The pin length was 4 mm, exceeding the thickness of the Al top plate, to induce sufficient mechanical stirring in these two base metals. Joining was performed with a tool rotation speed of 900 rpm to 1500 rpm (900 rpm, 1200 rpm, and 1500 rpm), a constant tool penetration depth of 0.3 mm, and a constant dwell time of 15 s.

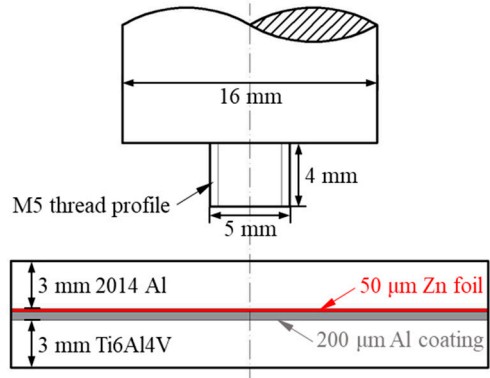

**Figure 2.** Schematic of the set-up used in the friction stir spot welding-brazing (FSSW-B) process.

To study the characteristics of the joint, the samples were sectioned transversely and polished for metallographic examinations. The microstructure and chemical composition of the cross-section of joints were examined using an FEI Inspect S50 scanning electron microscopy (SEM, ThermoFisher, Hillsboro, OR, USA) equipped with an Oxford Inca X-Act energy dispersive spectrometry (EDS, ThermoFisher, Hillsboro, OR, USA).

The tensile shear properties of the FSSW-B joint and FSSW joint at same welding conditions were tested to compare the mechanical properties. The dimension of the specimens was 125 mm in length and 35 mm in width. A supporting plate was placed at each end of the tensile specimen to maintain the joint weld region parallel to the tensile loading direction. The lap-shear tensile tests were conducted at

room temperature on a WDW-50 tensile test machine with a constant crosshead speed of 1 mm/min. The fracture surfaces after tensile testing were examined by SEM and EDS.

## 3. Results and Discussion

### 3.1. Macrostructure

The cross-section of the FSSW-B joint produced at a tool rotation speed of 1200 rpm is shown in Figure 3. Macroscopically, the Al coating is still visible at the interface, while the Zn interlayer is hard to observe. Retraction of the tool caused a visible keyhole at the center of the joint and a shallow indentation at the top of upper Al alloy. A partially bonded region (commonly referred to as the hook [37,38]) formed by upward bending of the interface is clearly observed as well. In a traditional FSSW joint, the distance from the tip of the hook to keyhole interface (including stir zone and hook region) is addressed as the bond width of weld due to the existence of an unbonded region outside of the hook [39]. In this study, another joining zone under the shoulder is detected outside of the hook as shown in Figure 3, which is named as the brazing zone and will be analyzed in detail later.

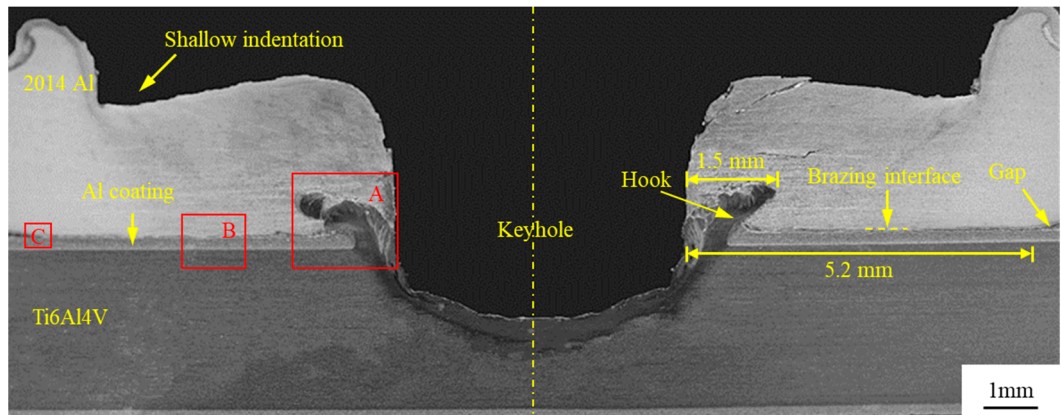

**Figure 3.** Typical cross-sectional of the FSSW-B joint made at the rotation speed of 1200 rpm. The labeled dimensions were measured via the optical microscope measurement.

### 3.2. Microstructure

Figure 4a shows the microstructure of the stir zone (region A marked in Figure 3) with several areas of interest (S1, S2, and S3) as highlighted by rectangular boxes. EDS element distribution maps of these selected regions in the stir zone are present in Figure 5. It indicates that the darker region consists of the Al-rich phase and the lighter region consists of the Ti-rich phase.

One of the important factors that can influence the strength of a dissimilar weld is the formation and distribution of IMCs [40–42]. The formation of IMCs during FSW of Al and Ti alloys has been discussed in detail elsewhere [7,43]. Generally, $TiAl_3$ IMC is preferable in FSWed Al/Ti joints because the $TiAl_3$ phase is the only transient phase when the reaction temperature is lower than the melting point of Al [44]. Thermodynamically, the free energy of formation of $TiAl_3$ is also the minimum among $Ti_3Al$, $TiAl$, and $TiAl_3$ compounds [8]. In the current study, $TiAl_3$ IMC has also been observed in the stir zone, which exists in two features, layer-like and particle-like. The layer-like $TiAl_3$ as marked in Figure 4b is about 10 μm in thickness, showing a mechanical interlocking with the base metals. It is formed because of the mixing of Al and Ti at a high temperature during friction stir processing [45]. Apparently, the particle-like $TiAl_3$ in the stir zone comes from original Al coating, which has been stirred into the Al alloy by the rotating pin. However, they aggregate at the interface outside the hook (see Figure 4c) and the tip region of the hook (see Figure 4d) caused by vigorous mixing and deformation. Interestingly, no Zn element was detected in these regions, indicating the added Zn interlayer has been removed from the stir zone (see Figure 5). Therefore, the Zn interlayer and Al coating have no significant influence in the stir zone of the FSSW-B joint.

Figure 6 displays the microstructure of a reaction layer with an average thickness of about 22 μm between the upper Al alloy and Al coating (region B marked in Figure 3). There are numerous TiAl$_3$ particles present in this reaction layer, and these particles maintain their original shape. EDS line scanning results reveal that the matrix of this reaction layer mainly consists of Al and Zn elements (see Figure 6b,c). Therefore, most of the TiAl3 particles are likely to remain intact and no obvious IMC is generated in the reaction layer during the FSSW-B process. According to the presence of Zn in this layer, it is reasonable to conclude that the Al alloy and Al coating were brazed with the Zn interlayer. Moreover, the thickness of the Al coating, about 200 μm (as shown in Figure 6a) in this brazing zone, is close to that before the joining process, suggesting that this FSSW-B process could not cause significant changes in the Al coating underneath the shoulder. At the end of the brazing zone (region C marked in Figure 3), as shown in Figure 6d, a large amount of lamellar and island Al–Zn eutectic structures, including the dark Al-rich phase and light Zn-rich phase, are observed. This confirms the melting of the Zn foil during the FSSW-B process [29].

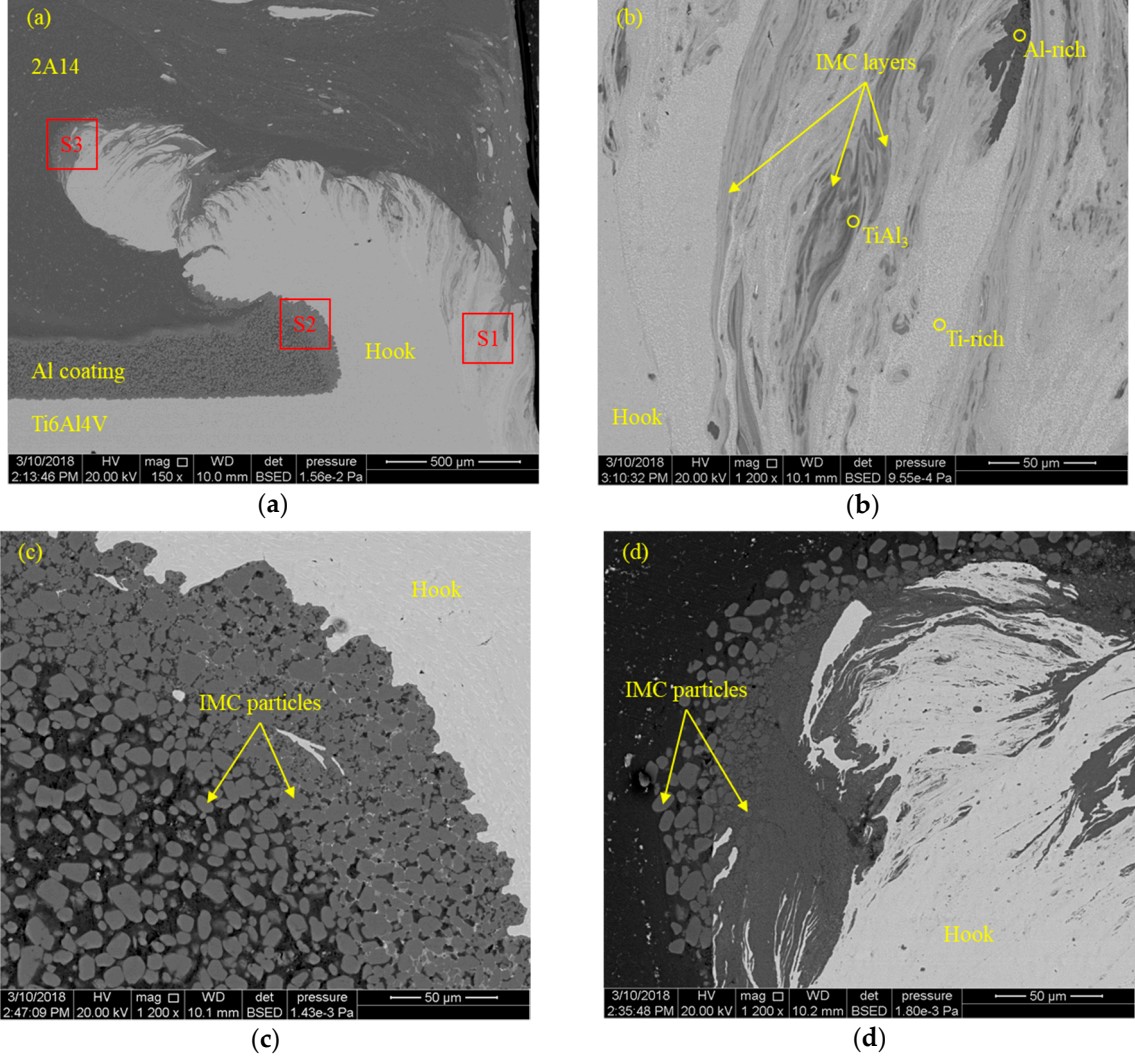

**Figure 4.** Scanning electron microscopy (SEM) image showing different regions in the stir zone of the FSSW-B joint; (**a**) high magnification view of the hook zone (region A marked in Figure 3a); (**b**) region S1 indicating the interlocking; (**c**) region S2 indicating the tip of the hook; and (**d**) magnified view of region S3. IMC—intermetallic compound.

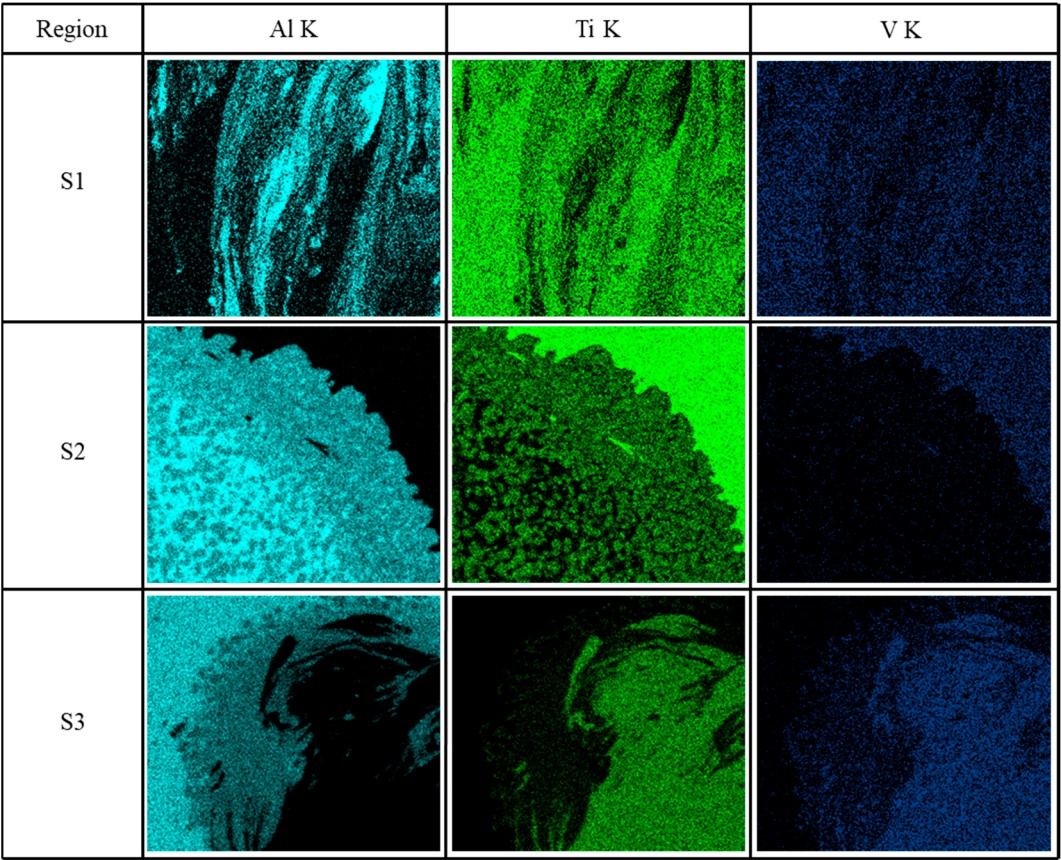

**Figure 5.** Energy dispersive spectrometry (EDS) element distribution maps of selected regions in the stir zone (red square marked in Figure 4a).

Herein, the formation process of the FSSW-B joint can be elucidated as follows: When the pin penetrates into the upper Al sheet, the temperature at the interface underneath the stirring pin rises rapidly with the gradual penetration as a result of the combined action of frictional heat and plastic deformation. The Zn interlayer would melt if the local temperature reaches its melting point. Meanwhile, the liquid Zn diffuses into the upper Al sheet and lower Al coating resulting, in the formation of the Al–Zn eutectic. However, when the stir pin comes into contact with the top surface of the lower Ti6Al4V sheet, most of the interfacial structure underneath the stirring pin starts to be destroyed as the tool penetrates further. At the same time, the heated and softened material adjacent to the pin deforms plastically, resulting in mixing and solid-state bonding between the upper Al and lower Ti alloys. When the shoulder comes into contact with and penetrates into the upper Al alloy, Zn foil under the shoulder melts and reacts with the upper Al sheet and lower Al coating, also resulting in the formation of the Al–Zn eutectic. Notably, the Al–Zn eutectics formed above are created along the interface between the upper Al sheet and Al coating. They can be squeezed into the gap out of the shoulder area as a result of extrusion force, which would help to remove the oxides from metal surfaces [22,46]. Finally, similar to the FSSB process [22,29,47], a brazing zone is formed underneath the shoulder. This brazing layer has a curved interface (see Figure 6a) because of the uneven forging force created by the shoulder [48].

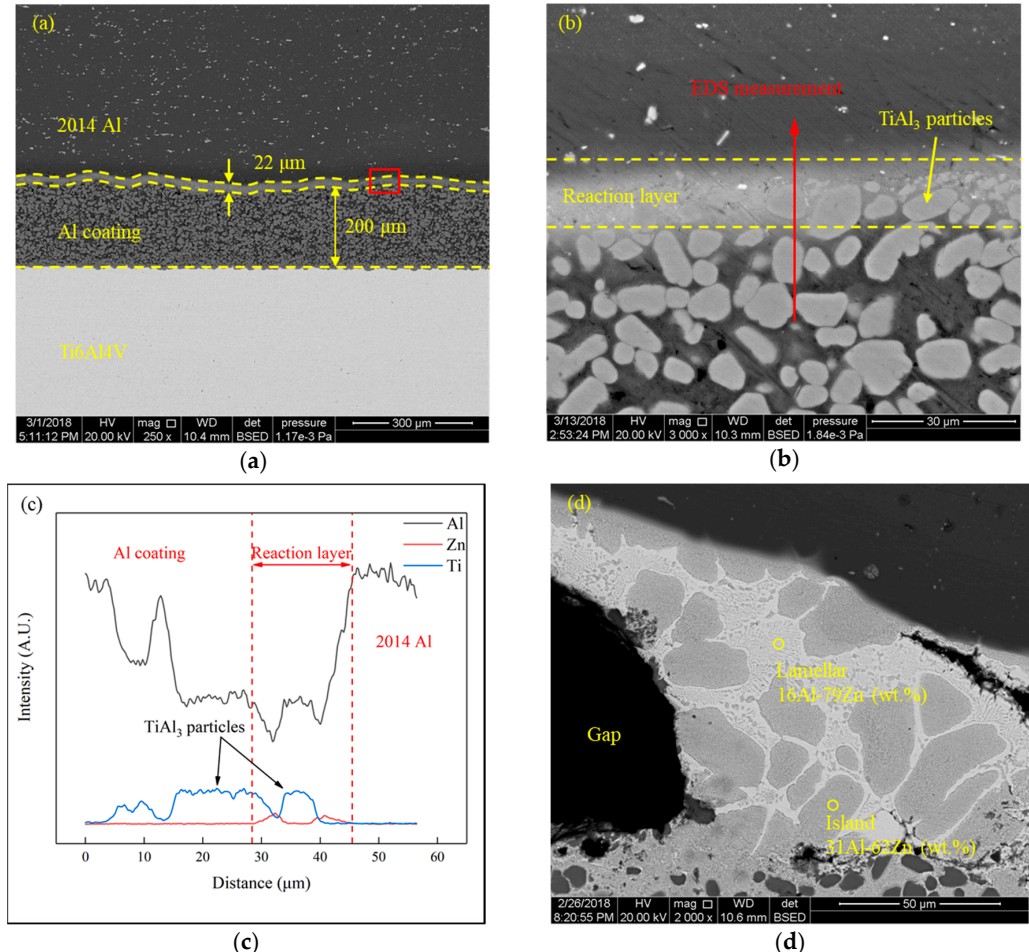

**Figure 6.** Microstructure of the brazing zone of the joint with (**a**) low magnification (region B marked in Figure 3a) and (**b**) high magnification (red square marked in (**a**)); (**c**) EDS results across the reaction layer (red line marked in (**b**)); (**d**) microstructure of the region at the end of the brazing zone (region C marked in Figure 3a), showing the Al–Zn eutectic structure. The thickness of the reaction layer was based on the measured distribution of Zn element.

## 3.3. Mechanical Property

Thanks to the formation of such a brazing zone, an enlarged bonding area, the lap shear fracture load of this FSSW-B joint is significantly increased compared with the FSSW joint. Figure 7a plots typical load–displacement curves of the joint produced by FSSW-B and FSSW at the same processing parameters (1200 rpm tool rotation speed). It can be seen that they both have no obvious yield platform, while the fracture load and fracture strain of FSSW-B joint are higher than those of the FSSW joint.

Figure 7b depicts the fracture surface (Ti6Al4V side) of the FSSW-B joint after the tensile shear test. It clearly reveals that the primary crack initiates from the brazing zone, then propagates through the stir zone, and finally through the keyhole to the other side. Meanwhile, as marked in Figure 7b, two identifiable regions are observed in the stir zone (region I and II) and brazing zone (region III and IV), respectively.

Figure 8 displays the details of all four regions of the fracture surface. It can be seen that all these four regions exhibit brittle fracture morphology, confirming the brittle fracture in the loading curve. The EDS analysis in region I (see Figure 8a) shows 70 wt.% Ti, indicating the separation of this region was mainly through the hook. TiAl3 particles can be observed in region II (see Figure 8b) and III (see Figure 8c), suggesting the separation was along the hook and through the reaction layer, respectively. The fracture surface of region IV is composed of about 36 at.% Al and 62 at.% Ti and appears to be flatter microscopically than other regions (see Figure 8d). It suggests that the fracture

was at the continuous TiAl$_3$ layer between Al coating and Ti6Al4V alloy. It is worth noting that the aluminizing could help with increasing the joining area, while TiAl$_3$ in this Al coating would affect the mechanical properties of the joint.

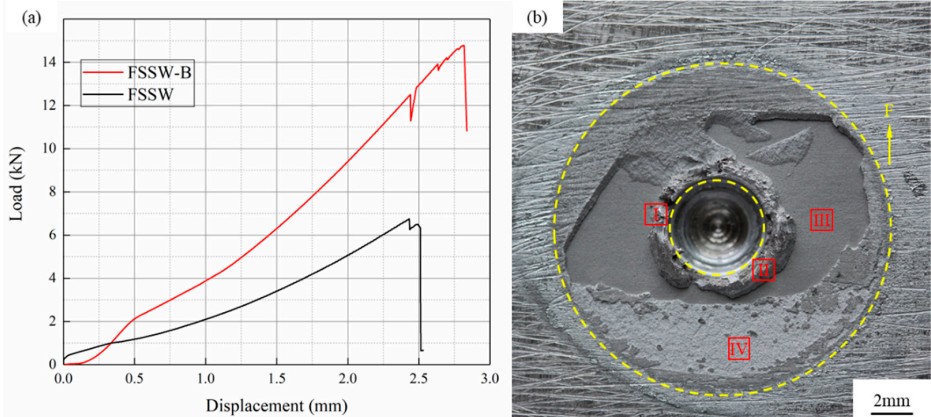

**Figure 7.** (**a**) Comparison of the lap shear fracture load of FSSW-B and FSSW joint at the same process parameter (1200 rpm); (**b**) macrograph of the fracture surface of FSSW-B joint (Ti6Al4V side). The yellow dotted lines in (**b**) mark the effective joining zone.

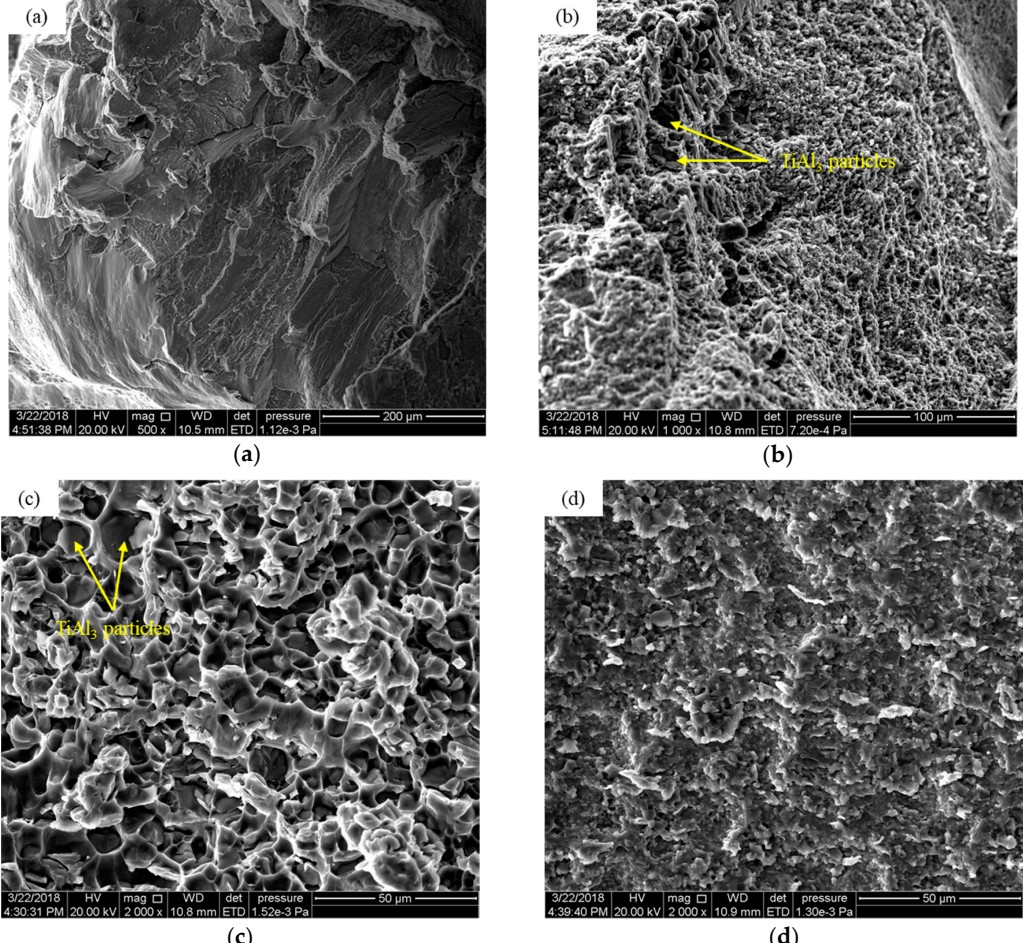

**Figure 8.** Highlighting the details of the fracture surface of each region marked with the red square in Figure 7b: (**a**) region I; (**b**) region II; (**c**) region III; and (**d**) region IV.

### 3.4. Parametric Study

Undoubtedly, the mechanical property of joints significantly depends on the process parameters. Generally, the heat input can be increased by increasing penetration force, angular velocity, and sampling time [49]. As the increase of penetration depth and rotation speed could increase the formation of IMCs [39,50], only the influence of rotation speed on the fracture load of joints was considered in the current study.

Figure 9 compares the fracture load of joints produced by FSSW-B and FSSW, showing that the change of failure load of FSSW-B joints is significantly greater than that of FSSW. At a low rotation speed, 900 rpm, their average fracture loads are quite close. The fracture load of the FSSW-B joint is much higher than that of the FSSW joint when the rotation speeds are 1200 rpm and 1500 rpm, respectively.

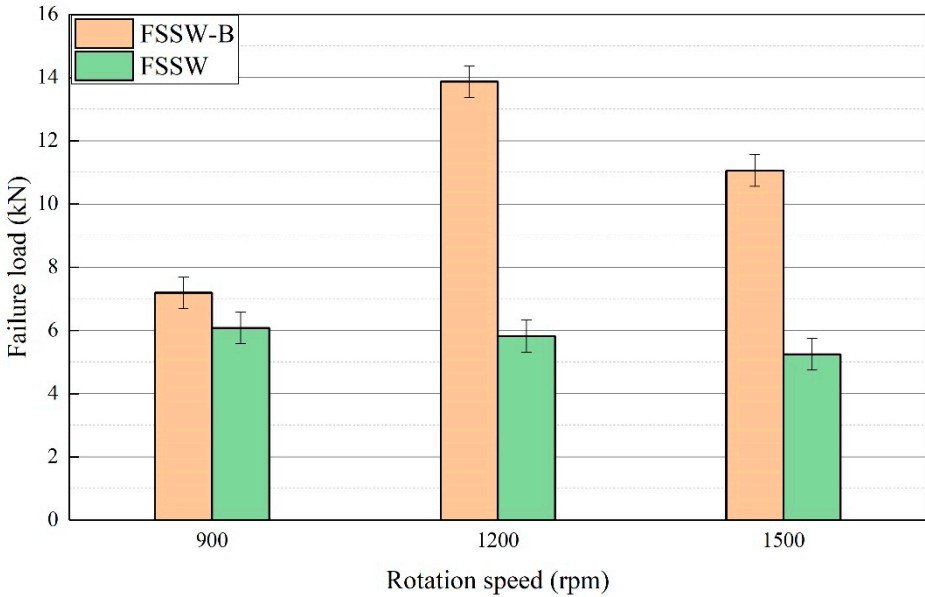

**Figure 9.** Comparison of lap shear fracture load of FSSW-B and FSSW joint at the same tool rotation speeds.

To study the variation in the fracture load, Figure 10 compares the microstructure of the brazing zone formed at the rotation speed of 900 rpm and 1500 rpm, respectively. As shown in Figure 10a, an unbonded region exists in the reaction layer because of insufficient heat input at a low rotation speed for the brazing process in this area, resulting in limited contribution to the fracture load increment. A continuous brazing zone is successfully fabricated as the rotation speed is increased to 1200 rpm and 1500 rpm. The thickness of reaction layer reduces from 22 μm (see Figure 6a) to 17 μm (see Figure 10b) because more liquid has been squeezed out of the brazing zone, caused by increased heat input, which is agreement with that of the Al/Cu joint made by pinless FSW using a Zn filler [51]. Therefore, sufficient energy input is necessary to ensure the successful brazing of Al alloy and Al coating, which play a key role in determining the joining width of the joint.

Similar to the traditional FSSW [37,39,52], the geometrical features (including indentation depth, hook height, and orientation) vary with rotation speed, as shown in Figure 11. When the tool rotation speed is 1500 rpm, compared with rotation speeds of 1200 rpm and 900 rpm, the hook terminates much closer to the keyhole. Although the tool penetration depth remains constant in the current study, more material is drawn out with the tool as the heat input increases [53,54]. This could cause the decreasing of the effective top sheet thickness. These varied geometrical features reduce the solid-state bonded width of the joint and thereby reduce the failure load of the FSSW joint from 6.6 kN to 4.3 kN (see Figure 9). Obviously, it would also affect the failure load of the FSSW-B joint.

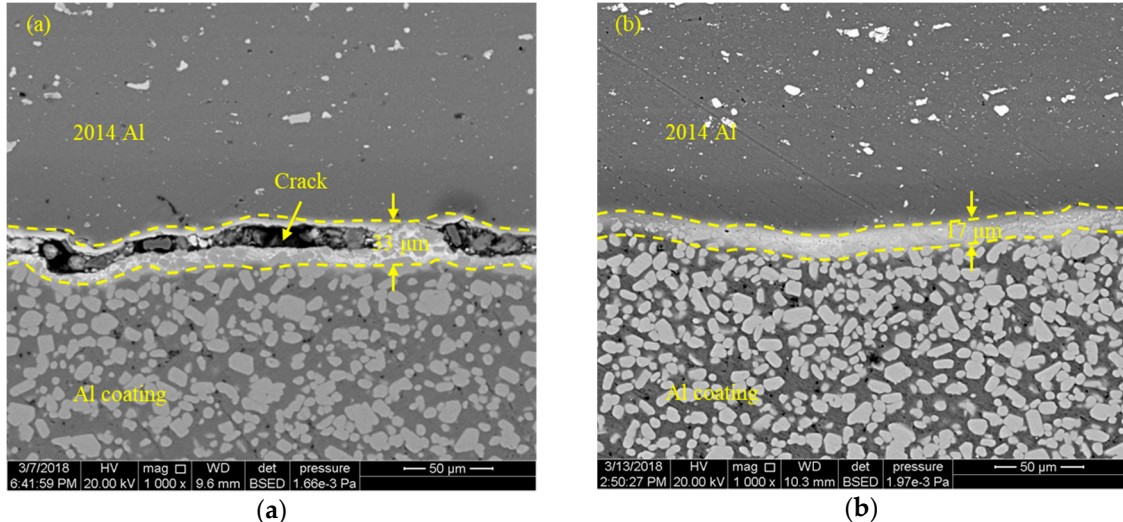

**Figure 10.** Microstructure of the brazing zone of the FSSW-B joint made at the tool rotation speed of (**a**) 900 rpm and (**b**) 1500 rpm.

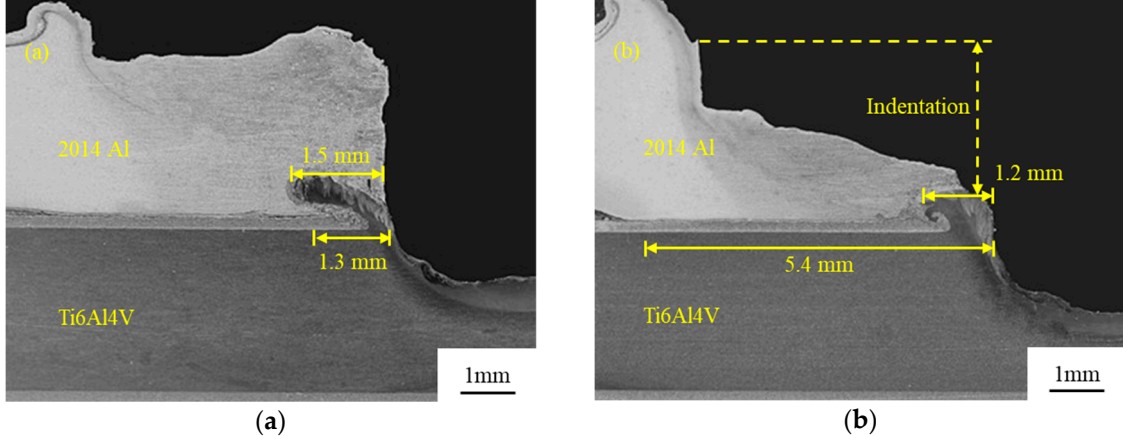

**Figure 11.** Microstructure of the stir zone of the FSSW-B joint made at the tool rotation speed of (**a**) 900 rpm and (**b**) 1500 rpm. The labeled dimensions were measured via the optical microscope measurement.

In the current study, the joining zone is a ring-like area that extends radially from the keyhole interface to the end of the brazing zone. Here, the diameter of this joining zone is described as the joining width. When the rotation speed is 900 rpm, the joining width is approximately 1.3 mm as measured in Figure 11a, which is smaller than the solid-state bonding width, 1.5 mm. This suggests that the brazing zone is almost negligible and does not contribute to the fracture load of the FSSW-B joint in comparison with the FSSW joint. Upon increasing the rotation speed to 1200 rpm, the joining width increases to approximately 5.2 mm (see Figure 3). Here, the sum of this joining width (5.2 mm) and the radius of keyhole (about 2.5 mm) is close to the radius of the used shoulder (8 mm) when the rotation is 1200 rpm. Therefore, this size might have reached the maximum value owing to the limited thermo-mechanically affected zone determined by the size of the shoulder. Therefore, as the rotation speed further increases to 1500 rpm, the joining width slightly increases to 5.4 mm (see Figure 11b). However, the reduction of solid-state bonding width, from 1.5 mm to 1.2 mm, results in the decreasing of joining width and thereby reduces the fracture load. This fact explains why the fracture load of the FSSW-B joint is first increased and then decreased with the increase of rotational speed.

## 4. Conclusions

This study provides an improved friction stir spot joining process for the joining of Al and Ti alloys. The following conclusions are derived:

- In addition to the solid-state joining of 2014-T4 Al and Ti6Al4V alloys via the FSSW technique, brazing between the Al alloy and Al coating on Ti6Al4V alloy was successfully introduced by the addition of a Zn interlayer.
- The Zn interlayer and Al coating have no significant influence in the stir zone of the FSSW-B joint. Because of the extrusion force during the joining process, the introduced TiAl$_3$ particles from Al coating are squeezed into the brazing zone, while the formed Zn–Al eutectic is squeezed out of the brazing zone.
- The formation of the brazing zone significantly increases the joining area, causing the highest fracture load of the FSSW-B joint to be improved by 110% compared with that of the traditional FSSW joint.
- Fracture load of the joint was first increased and then decreased with increasing of the rotational speed, which was rationalized to the varied effective joining areas.

**Author Contributions:** Investigation, X.Z., S.L., and K.H.; Resources, Y.H.; Supervision, Y.C.; Writing—original draft, X.Z.; Writing—review & editing, S.L. and P.P.

**Funding:** The research was financially supported by the National Natural Science Foundation of China (No. 51865035), the Fund for Jiangxi Distinguished Young Scholars (2018ACB21016), the Project for Jiangxi Advantageous Scientific and Technological Innovation Team (20171BCB24007 & 20181BCB19002), and Aviation Science Funds of China (2017ZE56010).

**Conflicts of Interest:** The authors declare no conflict of interest.

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
