# Peer review of "Friction Stir Spot Welding-Brazing of Al and Hot-Dip Aluminized Ti Alloy with Zn Interlayer"

_metals, doi:10.3390/met8110922_

Round 1

Reviewer 1 Report

Do all mandatory changes in attached file

Author Response

Thank reviewer for his/her comments for improvement of the manuscript. Responses could be found from the attachment.

Reviewer 2 Report

Very interesting modification to the FSSW technique that is well supported by various characterization methods. I would only recommend that the paper receive minor editing for English grammar.

Author Response

Response to the comments of Reviewer #1:

Very interesting modification to the FSSW technique that is well supported by various characterization methods. I would only recommend that the paper receive minor editing for English grammar.

Response: Thanks, we have revised the manuscript carefully and tried to avoid any grammar errors.

Reviewer 3 Report

Dear authors,

The work you present is interesting and deserves attention. In full agreement with you I also believe that the Friction Stir Spot Welding can be enhanced metallurgically in order to improve the bonding and to create higher strength joints. Thus, the support of the FSSW by brazing is interesting and can provide solutions in joints that usually are compromised by hard intermetallics. The methodology applied is clear, so are the results and conclusions. The manuscript can be published in its current form, but I urge you to provide an additional figure, this of the hardness in the joint area. Especially due to the occurrence of these mixed zones with the TiAl3 intermetallics, I would expect to observe the progress of the hardness. This would of course further support the good results achieved after the application of your process. A question, did you monitor the temperature evolution? This is of significance in order to better support the Zn diffusion path and to explain the reason for the occurred crack based on data. 

Herewith I accept you manuscript in its current form.

Sincerely,

The reviewer

Author Response

Thank reviewer for his/her comments for improvement of our manuscript. Responses could be found from the attachment.

Round 2

Reviewer 1 Report

Do as it was suggested previouly.
